# Targeted Approaches in Metastatic Castration-Resistant Prostate Cancer: Which Data?

**DOI:** 10.3390/cancers14174189

**Published:** 2022-08-29

**Authors:** Claudia Mosillo, Maria Letizia Calandrella, Claudia Caserta, Serena Macrini, Annalisa Guida, Grazia Sirgiovanni, Sergio Bracarda

**Affiliations:** Medical and Translational Oncology, Department of Oncology, Azienda Ospedaliera Santa Maria, 05100 Terni, Italy

**Keywords:** prostate cancer, target therapy, mCRPC, precision medicine, DDR, PTEN, MSI, dMMR

## Abstract

**Simple Summary:**

Castration-resistant prostate cancer (CRPC) remains an incurable disease, but some promising innovative treatment options are under investigation. Recent developments in precision medicine have enabled the identification of new predictive biomarkers and potential targeted agents. The purpose of this review is to summarize and discuss new therapeutic approaches for metastatic CRPC (mCRPC), focusing on pathway description, prognostic and/or predictive role of recently discovered molecular alterations, investigation techniques, and potential clinical implications.

**Abstract:**

Prostate cancer is the second most common diagnosed cancer and the fifth leading cause of cancer-related deaths in men worldwide. Despite significant advances in the management of castration-sensitive prostate cancer, the majority of patients develop a castration-resistant disease after a median duration of treatment of 18–48 months. The transition to a castrate resistance state could rely on alternative survival pathways, some related to androgen-independent mechanisms. Although several agents have been approved in this setting, metastatic castration-resistant prostate cancer (mCRPC) remains a lethal disease. Recent studies revealed some of the complex pathways underlying inherited and acquired mechanisms of resistance to available treatments. A better understanding of these pathways may lead to significant improvements in survival by providing innovative therapeutic targets. The present comprehensive review attempts to provide an overview of recent progress in novel targeted therapies and near-future directions.

## 1. Introduction

Prostate cancer is the second most diagnosed cancer and the fifth leading cause of cancer death in 2020 among men worldwide [1]. The androgen deprivation therapy (ADT) still represents the backbone of treatment for advanced diseases. Even though the systemic therapy of the metastatic castration-sensitive disease was intensified by the addition of one or two drugs (chemotherapy and/or androgen-receptor-targeting agents), about half of these patients develop a CRPC in approximately 5 years [2,3,4,5,6,7]. Despite significant advances in the last decade, mCRPC remains a lethal disease with a median survival of about 30 months [8]. Progression to the castration resistance state may involve androgen-dependent and -independent mechanisms. Recent studies have detailed these complex pathways and potentially actionable targets useful for the development of new therapeutic strategies [9]. A multi-institutional integrative clinical sequencing analysis reveals that approximately 90% of mCRPC harbor clinically actionable, somatic or germline, gene alterations. Most cases remain dependent on androgen receptor (AR) signaling finding aberrations in the AR gene in about 60% of the patients. Non-AR-related actionable alterations included aberrations in the PI3K pathway (49%), DNA repair pathway (19%), RAF kinases (3%), CDK inhibitors (7%), and the WNT pathway (5%) [10]. Moreover, a variable but low frequency (1–12%) of mismatch repair–defective (dMMR) has been reported in mCRPC in different studies [11]. Recent studies suggest that dMMR cancers may benefit more from immune checkpoint-inhibiting therapies than mismatch repair–proficient tumors (pMMR) [12]. Recently, several different novel treatment strategies have been shown to improve outcomes in advanced prostate cancer (Table 1). The aim of this review is to summarize and discuss the present results of new molecular targeted approaches for mCRPC.

## 2. New Therapeutic Approaches for Metastatic Castration-Resistant Prostate Cancer

### 2.1. DNA Damage Repair Pathways

In a normal cell, it is very important to preserve DNA integrity, and several molecular mechanisms are constantly active to repair endogenous and exogenous DNA damage (DNA Damage Response—DDR). The double-strand break (DSB) is a “bulky” DNA damage implicated in tumorigenesis and other human genetic disorders. Among various cellular reparative mechanisms, the homologous recombination repair (HRR) system is involved in this kind of DNA damage. The HRR mechanism includes a variety of proteins orchestrated in a process ensuring genomic stability and cell survival [13]. The breast cancer type 1 and 2 protein (BRCA 1/2) are the essential mediators of the HRR system, favoring the recruitment of other catalytic proteins (partner and localizer of BRCA2—PALB2; ataxia telangiectasia mutated—ATM; ataxia telangiectasia and rad3-related—ATR; DNA repair protein RAD50 and RAD51; Double-strand break repair protein MRE11; checkpoint kinase 2—CHEK2; X-ray repair cross complementing 2/3—XRCC2/3; histone H2AX; cellular tumor antigen p53), and leading to repair the DSB with high fidelity [14,15]. Consequently, inherited (germinal) and/or acquired (somatic) mutations in these pathways genes leads to the loss of the HRR mechanism resulting in the cell’s inability to repair DNA damage (homologous recombination deficiency—HRD), thus promoting tumorigenesis. Germinal alterations of these genes can define, with variable penetrance, an individual predisposition to the oncological disease. Contrarily, the acquired gene mutations are found only in tumor tissue and consist of a dynamic condition changing over time in response to therapies and genetic instability. About 27% of mCRPC patients, unselected for age at diagnosis or family history, harbored a germline or somatic alteration in a DDR gene, and BRCA1-2, ATM, and CHEK2 are the most frequently affected (Table 2) [16]. The germline pathogenic mutations are related to unfavorable histopathological features (high Gleason Score, intraductal/ductal histology, and lymphovascular invasion), high PSA level and/or advanced stage of disease at initial diagnosis, and a poor prognostic outcome [17]. To date, germline genetic testing, performed from a blood or saliva sample, is recommended for patients with localized or metastatic prostate cancer and a strong family history of malignancy or germline mutations. In patients with a localized tumor, this test provides only prognostic information, while a positive result in metastatic disease has today a predictive role. The somatic mutations analysis, through next-generation sequencing (NGS)-based approaches, can be performed on tumor sample sources such as biopsies, surgical material, circulating tumor DNA (ctDNA), and circulating tumor cells (CTCs). The NGS testing success rates depend on tumor sample characteristics, with better accuracy for newly collected metastatic samples or archival tissue aged < 1 year [18,19]. However, somatic mutations observed in tumor tissue may change over time due to genetic instability and selective pressure from therapy. Thus, repeat testing of tumor DNA may be appropriate during the disease course. Testing of tumor tissue is suggested in patients with mCRPC to provide relevant therapeutical information. Several pieces of evidence showed that an inefficient tumor DDR system can be exploited therapeutically using DNA-targeting therapeutic agents [20]. HRD-positive tumors are preferentially sensitive to poly (ADP-ribose) polymerase (PARP) enzymes inhibitors and to specific chemotherapy class drugs, such as alkylating agents (mainly carboplatin) or antimetabolites, producing great DNA damage and causing cell death directly or during DNA replication. The PARP enzymes represent a crucial system for repairing single-strand DNA damage, and the PARP inhibition leads to DSBs. Accordingly, when HRR is not able to correct this damage, the tumor cell does not survive. This therapeutic strategy, defined as “synthetic lethality”, is now the cornerstone of DDR pathway management in cancer treatment, with a consequent increasing use of PARP inhibitor monotherapy for the treatment of several solid tumors, including mCRPC (Table 3) [21]. In May 2020, the U.S. Food and Drug Administration (FDA) approved olaparib for the treatment of mCRPC with deleterious germline or somatic HRR gene mutations and disease progression during the treatment with at least enzalutamide or abiraterone. Olaparib is a multitarget PARP inhibitor (PARP1, PARP2, and PARP3 enzymes) with activity in ovarian cancer as well as other solid tumors. The TOPARP-A trial is the first phase II, single-arm study investigating olaparib in mCRPC. This study enrolled 49 patients not selected for mutation in DNA-repair genes and previously treated with docetaxel and novel hormonal agents. Results of this study demonstrated a promising antitumor activity with durable disease control, particularly in patients with HRD (ORR 33% in the overall population and 88% in HRD population—rPFS 0.7 months in the overall population and 9.8 months in HRD population) assuming a predictive role of these biomarkers [22]. Subsequently, the TOPARP-B phase II study randomized 98 patients with mCRPC and DDR gene mutations to receive olaparib 400 mg twice daily (TD) or olaparib 300 mg TD. The final data (median follow-up: 24.8 months) showed no significant differences in activity between the two study arms. The ORR was about 40%. Interestingly, the subgroup analyses by aberrant genes revealed higher response rates for BRCA 1/2 patients (83.3%) compared with other alterations (25% for ATM and 37% for PALB2) [23]. This evidence suggested a significant clinical benefit for olaparib only in a specific subset of patients with HRD. The latest prospective study published, which tested olaparib in patients with HRD, was the PROfound trial. This phase 3 study enrolled 386 patients in two different cohorts: cohort A (*n* = 245), including patients with BRCA1, BRCA2, or ATM gene mutations, and the cohort B (*n* = 142), enrolling patients with other 12 genes alterations, such as BRIP1, BARD1, CDK12, CHEK1, CHEK2, FANCL, PALB1, RAD51, RAD54, and PPP2R2A. In each cohort, the population was randomly assigned in a 2:1 ratio to olaparib 300 mg TD or to a novel hormone therapy (enzalutamide or abiraterone) and was stratified according to previous taxane therapy or not and the evidence or not of measurable disease. A crossover from the control arm to olaparib was allowed in case of progression. The primary endpoint was radiological PFS (rPFS) in cohort A, with the secondary endpoints (ORR in cohort A, rPFS in the overall population, time to pain progression, and OS in cohort A) to be performed in a hierarchical manner if the primary endpoint was met. A statistically significant difference in rPFS was observed in the olaparib group versus control group in cohort A (7.4 vs. 3.6 months; HR 0.34; 95% CI, 0.25–0.47; *p* < 0.001). In the same cohort, a higher ORR and OS was reported in favor of the olaparib arm (ORR 33% vs. 2%, *p* < 0.001—OS 19.1 vs. 14.7 months, *p* = 0.02). Moreover, a sensitivity analysis for OS with adjustment for crossover (80% of cases) showed an HR of 0.42 (95% CI, 0.19–0.91) [24]. Consistent with previous phase 2 studies, the subgroup analysis by mutation showed a clear advantage for BRCA ½ deficient patients compared with ATM or other carriers in terms of rPFS. Considering these data, the FDA approved a genomic test for a better selection of patients with mCRPC carrying BRCA1/2 alterations. In the same year, the FDA granted accelerated approval to a second PARP inhibitor, rucaparib, for the treatment of mCRPC patients with BRCA ½ mutation (germline and/or somatic) treated with 1–2 lines of androgen-receptor-directed therapy and a taxane. The efficacy of rucaparib was investigated in phase II, single-arm, ongoing TRITON 2 study. Eligible patients harboring germline and/or somatic alteration in BRCA ½ or other DDR pathways genes (e.g., ATM, FANCA, CHEK2, PALB2, BRIP1, CDK12, NBN, RAD51, and RAD54), received rucaparib 600 mg TD. The preliminary results concern efficacy and safety for patients with a deleterious BRCA ½ alteration. After a median follow-up of 17 months, the ORR by blinded independent radiology review of the evaluable patient’ population was 43.5% (43.5% (95% CI, 31–56.7%). Although the OS data were not mature at the time of the last analysis, the estimated 12-month OS was 73% [25]. The available study results allowed us to accurately identify patients who could benefit from PARP inhibitors. The ongoing phase 3 TRITON 3 trial is a randomized, open-label study evaluating rucaparib compared with physician’s choice drug (abiraterone, enzalutamide, or docetaxel) in mCRPC patients with specific gene alterations, including BRCA 1/2 or ATM (NCT02975934) who progressed after at least one prior androgen-receptor-targeted therapy and have not received prior chemotherapy. The other PARP inhibitors currently under investigation are talazoparib and niraparib. TALAPRO-1, a phase II single-arm study, recruited mCRPC patients with DDR who previously received at least one androgen-receptor-targeted therapy (enzalutamide and/or abiraterone) and docetaxel for castration-resistant disease. The study enrolled 128 patients who received talazoparib 1 mg once daily (OD) with ORR as the primary endpoint. After a median follow-up of 16.4 months, ORR was 29.8% (95% CI, 21.2–39.6%), while PFS was 5.6 months. Subgroup analysis showed a median PFS of 11.2 months in patients positive for BRCA1 or BRCA2 and a median PFS of 3.5 months only in those of the ATM group [26]. GALAHAD is another ongoing, single-arm, phase II study testing the PARP inhibitor niraparib. As of May 2019, 165 patients with pretreated mCRPC were recruited, of whom 81 patients had biallelic DDR gene alterations: 46 patients had BRCA 1 or 2 mutation, and 35 had non-BRCA alterations (e.g., ATM, FANCA, PALB2, CHEK2, BRIP1, or HDAC2). All patients received niraparib at a dosage of 300 mg OD. The preliminary results concerned the biallelic BRCA and non-BRCA population. In the BRCA group, ORR was 41%, with an rPFS and OS of 8.2 and 12.6 months, respectively [27]. PARP inhibitors are generally well tolerated; the most reported adverse events in patients with mCRPC were hematological toxicities (anemia, thrombocytopenia, and neutropenia) and gastrointestinal disorders (nausea and loss of appetite), largely managed with supportive care and dose modifications [23,24,25,26,27]. PARP inhibition represents an exciting tool for the management of HRD-positive mCRPC patients, and with the larger adoption of NGS technologies, the identification of these patients is likely to increase. Ongoing and future studies will be critical for an optimal understanding of the appropriate timing of PARP inhibition in mCRPC. Open questions remain regarding the clinical significance of monoallelic versus biallelic HRD status, not differentiated in most of the trials, and the relevance of germline compared to somatic-only mutations. Most recently, PARP inhibitors have been investigated in combination with androgen-receptor-signaling inhibition agents (abiraterone or enzalutamide) and immune checkpoint inhibitors, with the aim to improve disease response and patients outcome. Crosstalk between the DDR and androgen receptor signaling has been postulated, opening a new array of possible therapeutic strategies. According to preclinical studies, anti-androgen treatments may induce a BRCAness phenotype, which can be targeted by PARP inhibition. Some clinical trials support these results regardless of the HR status (Table 4).

### 2.2. MSI-H/dMMR Pathway

Microsatellite instability (MSI) is characterized by mutations in repetitive DNA sequence tracts caused by a failure of the DNA mismatch repair (dMMR) system. A deficient mismatch repair (dMMR) condition results from a biallelic mutational inactivation or epigenetic silencing of one or more genes of the dMMR pathway (most commonly MSH2, MSH6, MLH1, and PMS2). Due to a high tumor neoantigen burden, patients with MSI high (MSI-H) or dMMR tumors often exhibit greater and more durable responses to immune checkpoint inhibitor (ICI) treatment, regardless of the site of origin. In May 2017, the FDA granted accelerated approval of pembrolizumab, an anti-programmed cell death protein 1 (PD-1) antibody, for the treatment of MSI-H or dMMR solid tumors, after progression on at least one standard therapy. Historically, MSI was tested on tissue biopsies with PCR-based amplification followed by capillary electrophoresis, and more recently with next-generation sequencing (NGS)-based approaches. Different assays, using tumor tissue or plasma cell-free DNA (cfDNA) (liquid biopsies), are able to assess MSI status. MSI has been found in many cancer types, including colorectal (up to 15–20% of cases), endometrial (26–33%), ovarian (10%), cervical (8%), and gastric (8–22%) cancers. In prostate cancer, MSI-H and dMMR have been reported in a subset of tumors ranging from 1% in localized tumors up to 12% in metastatic disease [28,29]. In the analysis published by Nava Rodrigues et al. in 2018, the dMMR status was determined by loss of expression of the dMMR protein on IHC or evidence of MSI by PCR in 127 biopsies deriving from a cohort of 124 prostate cancer patients of the Royal Marsden Hospital. MSI was then evaluated in the same cohort with a targeted NGS panel (MSI-NGS). Overall, 10 patients (8.1%) had evidence of dMMR by IHC and/or MSI. Higher MSINGS score, used to score samples for an MSI-like phenotype by assessing targeted next-generation DNA sequencing data, was associated with dMMR, dMMR mutational signatures also associated with MMR gene mutations and increased immune cell, immune checkpoint, and T-cell-associated transcripts, including PD-L1, PD-L2, and PIK3CG. In a study reported in JAMA Oncology, Abida et al. describe a percentage of approximately 3% of patients with prostate cancer with MSI-H/dMMR tumors. The same patients exhibited durable responses to treatment with immune checkpoint inhibitors. This study evaluated 1551 tumor samples from 1346 prostate cancer patients treated at Memorial Sloan Kettering Cancer Center from January 2015 through January 2018. Tumor mutational burden (TMB) and MSI sensor score (quantitative measure of MSI) were assessed. About 3% of patients with evaluable disease had evidence of MSI-H or dMMR. A pathogenic germline mutation in a Lynch syndrome-associated gene was identified in 7/32 patients (21.9%), suggesting that germline testing should be considered for all patients with MSI-H/dMMR prostate cancer. Moreover, among the six patients with more than one tumor tissue analyzed, two patients exhibited an acquired MSI-H phenotype. These data suggest that metastatic tissue represents the optimal material for testing MSI status. Anti–PD-1/PD-L1 treatment was given to 11 MSI-H/dMMR mCRPC patients. Six of these patients had a significant decline in PSA value (more than a 50% decline compared to baseline), with four cases achieving a radiographic response. The authors concluded that the MSI-H/dMMR molecular phenotype, an uncommon but therapeutically meaningful state in prostate cancer, may be somatically acquired during disease evolution. Given the observed response to anti–PD-1/PD-L1 therapies, these findings support a prospective tumor sequencing for MSI-H/dMMR in all patients with advanced prostate cancer. However, since approximately half of the patients with MSI-H/dMMR had no response to immunotherapy, future studies should also explore the eventual mechanisms of resistance present in this population. Mechanisms may involve alterations in the tumor-antigen-presenting machinery and tumor-extrinsic factors, including inadequate T-cell activation [11]. At the 2021 American Urological Association Annual Meeting, Lenis et al. presented the results of a retrospective analysis of 2813 prostate cancer patients enrolled in the phase 3 IMPACT trial (NCT00065442). In this analysis, investigators evaluated the prevalence of MSI-H/dMMR and Tumor Mutational Burden-high (TMB-H). To meet the criteria for genomically identified MSI-H/dMMR, tumors needed to have an MSIsensor score of 10 or greater, or a score between 3 and 10 but harboring a deleterious alteration in MSH2, MSH6, MLH1, or PMS2. TMB-H tumors needed to have at least 10 mutations/megabase. In total, 64 patients (2.9%) were deemed to have MSI-H/dMMR tumors, 32 (1.4%) had TMB-H tumors, and 2146 (95.9%) had MSS/TMB-low (TMB-L) tumors. Across all included patients, the median age was 62 years (range, 56–68), and the patients had mainly high-grade tumors. Furthermore, 65.9% of patients had stage N0M0 disease, followed by those with M1 (27.5%) and N1M0 (6.5%) stage disease. This study showed that patients with MSI-H or TMB-H tumors were more likely to present a grade group 5 disease (56.2% and 53.1%, respectively) compared with MSS disease (34%). N1M0 and M1 disease stages were more prevalent in MSI-H (9.4% and 34.4%, respectively) and TMB-H tumors (15.6% and 18.8%) than in those with MSS and TMB-L tumors (6.2% and 27.4%, respectively). Moreover, of 21 MSI-H/dMMR patients treated with immunotherapy, 9 patients achieved a partial response, and 10 patients had stable disease as the best response. Of seven patients with TMB-H, five cases achieved stable disease, and two progressed. When comparing responses between the MSI-H/dMMR and TMB-H/MSS subgroups, data showed that only patients in the MSI-H/dMMR cohort experienced an objective response to immune checkpoint blockade. Additionally, the median radiographic progression-free survival was 41 months in the MSI-H/dMMR cohort compared with 7 months for patients with TMB-H disease (*p* < 0.01). The authors concluded that despite the limited sample size in the analysis, MSI-H/dMMR prostate cancers more frequently respond to immunotherapy compared with TMB-H prostate cancers [30]. Recently, Barata et al. published a multi-institutional, observational, prospective trial enrolling patients with advanced prostate cancer and MSI-H identified with circulating tumor DNA (ctDNA) NGS assay; 460 patients were screened, and MSI-H status was identified in 15 cases (3.7%). This trial represents the first experience in evaluating the predictive role of MSI-H status on ctDNA. The final analysis included 14/15 patients. Nine patients had mCRPC and received pembrolizumab: four patients achieved a PSA decline of ≥50% from baseline, including three patients with >99% PSA decline. Among patients evaluable for radiographic response (*n* = 5), the overall response rate was 60%, with one complete response and two partial responders [31]. However, to date, immunotherapy has been shown to have modest activity in prostate cancer. Studies conducted with single agents directed against PD-1, PD-L1, and CTLA4, have yielded substantially negative results with the low overall response. Moreover, the largest phase 2 trial with anti-CTLA-4 plus anti-PD-1 in mCRPC (CheckMate 650) showed positive preliminary results (ORR 25% in the pre-chemotherapy setting and 10% in the post-chemotherapy setting) but clinically significant toxicity (about 50% of grade 3–4 treatment-related adverse events and four treatment-related deaths) [32]. However, as a subset of patients may have durable clinical benefits with ICI, further studies are warranted to identify specific biomarkers (e.g., biallelic loss of CDK12) [33,34,35] and to increase the number of responders, also investigating combination strategies with second-generation hormonal therapies, chemotherapy, or PARP inhibitors (Table 5).

### 2.3. PI3K/AKT/mTOR Pathway

Phosphoinositine 3-kinases (PI3K)/RAC-α Protein kinase B (AKT)/mammalian target of rapamycin (mTOR) pathway is a set of signaling enzymes involved in intracellular signal transduction. One of its functions is to support cell growth, proliferation, motility, and survival. The phosphatase and tensin homolog (PTEN) is a phosphate protein involved in the regulation of the cell cycle and encoded by an onco-suppressor gene on chromosome 10. PTEN, by antagonizing the signaling of PI3K, negatively regulates the AKT/mTOR signaling cascade [36,37]. The loss of PTEN, resulting in tumorigenesis or drug resistance due to a hyperactivation of the PI3K/Akt/mTOR pathway, is the second most common genomic aberration in advanced prostate cancer after the androgen-receptor (AR) alterations [38]. This condition is described in about 15–20% of localized tumors and in 40–70% of mCRPC, more often caused by a gene deletion [39]. Immunohistochemistry (IHC) and fluorescence in situ hybridization (FISH) have been used to assess PTEN loss status in tumor tissue. Of note, in addition to gene deletion, the PTEN protein level may also be regulated by epigenetic silencing, making a PTEN status detection by FISH very difficult, while a direct detection by IHC of a cytoplasmic and nuclear PTEN loss is possible. Moreover, the IHC assay is less expensive and less time-consuming compared to FISH [40,41]. Over the past decades, some small molecule inhibitors of the AKT/PI3K/mTOR pathway have been investigated in prostate cancer. Early attempts to inhibit this signaling pathway concerned the inhibition of mTOR and PI3K. However, no significant antineoplastic activity has been demonstrated with single agents (e.g., everolimus and temsirolimus mTOR inhibitors or dactolisib and buparlisib PI3K inhibitors), probably due to a series of molecular events leading to an upregulation of AKT. Combination therapy resulted in significant toxicity and limited efficacy [42,43,44]. Over the years, AKT inhibition has proved to be the most promising strategy. Ipatasertib is an oral ATP-competitive inhibitor of AKT, and its activity is associated with high levels of AKT, loss of PTEN proteins, and PIK3CA kinase domain mutations [45,46]. Two phase I studies with Ipatasertib showed an acceptable tolerability profile characterized by gastrointestinal effects, asthenia/fatigue, hyperglycemia, rash, and preliminary anticancer activity in prostate cancer [46]. The subsequent, randomized phase II study evaluated the activity of abiraterone acetate (AA) in combination with Ipatasertib compared to abiraterone alone. This trial enrolled patients with mCRPC, unselected for PTEN loss, previously treated with docetaxel, and progressing after ≥1 hormonal therapy. A total of 253 patients were randomized (1:1:1) to receive AA 1000 mg OD and prednisone (PDN) 5 mg TD in combination with ipatasertib 400 mg OD or ipatasertib 200 mg OD, or AA and PDN plus placebo. All the patients were evaluated for PTEN status by IHC, FISH, and NGS. About the results, the safety of this drug combination was consistent with the previous clinical experience with ipatasertib alone. The adverse events were largely graded 1/2, easily manageable, and did not impact the dose intensity of both drugs. Moreover, ipatasertib plus AA and PDN demonstrated to prolong radiographic progression-free survival (rPFS) over placebo plus AA and PDN. The efficacy of the combination was greater in patients with PTEN loss compared to PTEN non-loss population and in patients with PTEN non-loss, but evidence of AKT hyperactivation due to other molecular aberrations detected by NGS (15–20%) [47]. Positive results of this study allowed the start of a phase III trial (IPATential150), in which enrolled patients were randomly assigned in a 1:1 ratio to AA plus ipatasertib 400 mg or AA plus placebo. All the enrolled patients who had asymptomatic or slightly symptomatic mCRPC were not pretreated for the castration-resistant setting and were unselected for PTEN expression. Stratification factors were taxane-based chemotherapy for hormone-sensitive prostate cancer, type of progression (radiological or biochemical only), presence of visceral metastases, and PTEN status by IHC. The study included two co-primary endpoints: rPFS in the intention to treat (ITT) population and rPFS in the PTEN loss population. In this trial, PTEN loss was defined as no detectable PTEN staining in PTEN loss patients (defined according to IHC assay demonstrating PTEN loss in ≥50% of the tumor cells). Out of 1101 enrolled patients, 521 (47%) were defined as PTEN loss. At data cutoff, with a median follow-up of 19 months, the trial results confirmed the efficacy of the combination over AA alone plus Prednisone (median rPFS 18.5 vs. 16.5 months, HR 0.77; 95% CI, 0.61–0.98, *p* = 0.034, significant at α = 0.04). However, any statistically significant benefit of the experimental approach was observed in the ITT population. Although overall survival was a secondary endpoint, this result was not yet mature [48]. Finally, an interesting exploratory analysis revealed that the magnitude of benefit with ipatasertib was more pronounced in patients with PTEN-loss tumors and alterations of the PIK3CA/AKT1/PTEN pathway, both evaluated by NGS [49]. These results support the idea that a more-stringent biomarkers analysis can be helpful in selecting the category of patients who can most benefit from ipatasertib plus abiraterone.

## 3. Potential Further Novel Agents for the Management of Metastatic Castration-Resistant Prostate Cancer

Novel pathways which act independently of the androgen axis are also being discovered; there are ancillary pathways such as RAS/MAP kinase, TGF-beta/SMAD pathway, FGF signaling, JAK/STAT pathway, Wnt-Beta catenin, and hedgehog signaling, which are involved in non-androgen-mediated mechanisms of resistance. Activation of these pathways in patients treated with AR-directed therapies led phenotypically to a process of epithelial–mesenchymal transition (EMT), a significant contributor to prostate cancer progression, development of metastasis, and therapeutic resistance through autocrine or paracrine mechanisms [50]. Molecules acting on these pathways are under investigation in phase I and II studies (Table 6). Recently, preclinical data provided evidence about the role of autophagy pathways in castration resistance, and it showed that the pharmacological regulation of this phenomenon can enhance the effectiveness of therapy in the mCRPC setting [51]. However, many of these pathways are extremely difficult to target because of the multitude of receptors, ligands, and downstream mechanisms involved. This makes the tested investigational agents not very manageable drugs, with significant toxicities. PSMA (prostate-specific membrane antigen), a transmembrane glycoprotein, is a well-characterized therapeutic and diagnostic target. A PSMA antibody-drug conjugate (ADC), a fully human IgG1 monoclonal antibody conjugated to the microtubule disrupting agent monomethyl auristatin E (MMAE), has demonstrated antitumor activity in pretreated mCRPC population; based on this antitumor activity in preclinical models and a phase 1 study, a phase 2 trial has been conducted in subjects who progressed following abiraterone/enzalutamide therapy. PSMA ADC, in a population of 119 subjects, showed modest activity in terms of PSA decline despite significant treatment-related AEs, including neutropenia and neuropathy with premature discontinuation [52]. These results have shown the importance of preliminary dose optimization, proper patient selection, and further evaluation in combination with other agents. Because of the recent development of efficacious and safe radio ligand approaches targeting PSMA and showing advantages in overall survival (177Lu/617PSMA), further development of this class of agent is doubtful [53,54].

## 4. Conclusions

New treatment options continue to enrich the therapeutic treatment scenario of mCRPC, thanks also to the introduction of precision medicine approaches. A better understanding of the molecular mechanisms and their possible interaction remains essential to optimize the use of available and future efficacious treatment options. Further explorations are needed to standardize approaches of gene sequencing and reporting of NGS analyses focused on the prostatic disease to optimize the selection of the right therapy at the right time for any individual patient and its eventual use in combination, a main clinical challenge in such a relevant disease.

## Figures and Tables

**Table 1 cancers-14-04189-t001:** Novel targeted therapies.

Molecular Alteration	Frequency in mCRPC	Therapy under Investigation
DDR pathways	27%	PARP inhibitors
MSI-H/dMMR pathway	1–12%	Immune checkpoint inhibitors
PI3K/AKT/mTOR pathway	49%	AKT inhibitors

DDR: DNA damage response; MSI-H: microsatellite-instability-high; dMMR: mismatch repair–defective.

**Table 2 cancers-14-04189-t002:** Prevalence of DDR gene mutations in mCRPC.

Rate of Germline and/or Somatic Mutations	Gene	Somatic	Germline
27%	BRCA2	7.7%	8.6%
	BRCA1	0.9%	0.9%
	ATM	4.5%	2.3%
	CHECK2	0.9%	4.1%

DDR: DNA damage response; mCRPC: metastatic castration-resistant prostate cancer.

**Table 3 cancers-14-04189-t003:** Phase II–III trials of PARP inhibitor in monotherapy.

Trial (Phase)	Enrolled Population	Selection for HRD	HRD Pts	Treatment Arms	Study Results
TOPARP-A (phase 2)	mCRPC; PD after docetaxel	No *	16/50 7 BRCA2 5 ATM 2 CHEK2 1 BRCA1 1 PALB2	Olaparib 400 mg TD	CR: 33% in all pts 88% in HRD pts
TOPARP-B (phase 2)	mCRPC; PD after docetaxel	Yes *	98 32 BRCA1/2 21 ATM 21 CDK12 7 PALB2 21 other	Olaparib 400 mg TD vs. Olaparib 300 mg TD	CR: 54.3% vs. 39.1%
TRITON 2 (phase 2)	mCRPC; PD after 1–2 ARTA and docetaxel	Yes	Cohort 1 13 BRCA1 102 BRCA2 Cohort 2 78 other **	Rucaparib 600 mg TD	Cohort 1 ORR: 43.5% Cohort 2 ORR:
TALAPRO-1 (phase 2)	mCRPC; PD after ARTA and docetaxel	Yes	75	Talazoparib 1 mg OD	ORR: BRCA1/2 43.9% PALB2 33.3% ATM 11.8%
GALAHAD (phase 2)	mCRPC; PD after ARTA and docetaxel	Yes	81	Niraparib 300 mg OD	ORR: BRCA1/2 41% Non-BRCA 9%
PROFOUND (phase 3)	mCRPC; PD after ARTA +/− docetaxel	Yes	Cohort A BRCA1 BRCA2 ATM Cohort B Other ***	Olaparib 300 mg TD vs. Enza 160 mg OD or Abi 1000 mg OD	Cohort A mPFS: 7.4 vs. 3.6 mo HR 0.34 *p* < 0.001

HRD: homologous recombination deficiency; Pts: patients; mCRPC: metastatic castration-resistant prostate cance; PD: progression disease; ARTA: androgen-receptor-targeted agents; TD: twice daily; OD: once daily; CR: composite response; Enza: enzalutamide; Abi: abiraterone; ORR: overall response rate; mPFS: median progression-free survival. * DNA sequencing was conducted using 113-gene panel including genes associated with PARP inhibition sensitivity and specifically genes involved in DNA repair. ** ATM, FANCA, CHEK2, PALB2, BRIP1, CDK12, NBN, RAD51, and RAD54. *** BRIP1, BARD1, CDK12, CHEK1, CHEK2, FANCL, PALB1, RAD51, RAD54, and PPP2R2A.

**Table 4 cancers-14-04189-t004:** Ongoing phase III trials of PARP inhibitor in combination with novel hormonal agent (NHA).

Combination Strategy	Population	Trial Identification
Olaparib + Abiraterone Acetate	Untreated mCRPC patients (docetaxel and NHA—not Abiraterone Acetate—in mHSPC are allowed), unselected for HRD	NCT03732820 (PROpel)
Niraparib + Abiraterone Acetate	Untreated mCRPC patients (docetaxel and NHA—not Abiraterone Acetate—in mHSPC are allowed), selected and unselected for HRD (2 cohorts)	NCT03748641 (MAGNITUDE)
Talazoparib + Enzalutamide	Untreated mCRPC patients (docetaxel and NHA—not Enzalutamide—in mHSPC are allowed), unselected for HRD	NCT03395197 (TALAPRO-2)
Rucaparib + Enzalutamide	Untreated mCRPC patients (docetaxel and NHA—not Enzalutamide—in mHSPC are allowed), unselected for HRD	NCT04455750 (CASPAR)

mCRPC: metastatic castration-resistant prostate cancer; NHA: novel hormonal agent; mHSPC: metastatic hormone-sensitive prostate cancer; HRD: homologous recombination deficiency.

**Table 5 cancers-14-04189-t005:** Ongoing phase III trials of ICI in combination with other therapies.

Combination Strategy	Population	Trial Identification
Pembrolizumab + Docetaxel	mCRPC patients previously treated with an NHA	NCT03834506
Pembrolizumab + Enzalutamide	Chemotherapy naïve mCRPC cases	NCT03834493
Atezolizumab + Enzalutamide	mCRPC patients previously treated with an NHA and Docetaxel	NCT03016312
Pembrolizumab + Olaparib	mCRPC after prior docetaxel and one NHA	NCT03834519

mCRPC: metastatic castration-resistant prostate cancer; NHA: next-generation hormonal agent.

**Table 6 cancers-14-04189-t006:** Potential novel agents.

Novel Pathway	Frequency	Strategy	Comments
WNT pathway	Wnt-activating mutations are observed in up to 20% of CRPC	-Inhibition of β-Catenin-Inhibition of Wnt Ligand Secretion	-Preclinical and phase I trials -Potential antitumor activity but significant toxicities -How and when assess alterations in Wnt signaling?
FGF pathway	FGFR1 was amplified in 10% of mPC	-Dovitinib (pan-class inhibitor including FGFR1, FGFR3, VEGFR1-3, PDGFRβ, fms-related tyrosine kinase-3, and c-KIT) -Erdafitinib	-Phase II trials ongoing -Modest antitumor activity -Not perform IHC staining or gene sequencing analysis of cancer tissue -Dose-limiting “off-target” toxicities
CDK4/6	Amplification of CDKN2A/B, CDKN1B, and CDK4 are observed in 5% of mPC	Cyclin-dependent Kinase 4/6 Inhibitor	-Phase Ib/II trials -Palbociclib did not impact outcome in RB-intact mHSPC
Ras–Raf–MEK–ERK Axis	Amplification of members within the MAPK pathway is as high as 32% in patients with mCRPC	EK1/2 inhibitors	Phase II trials ongoing
TGF-β pathway		TGF-β receptor I (TRI) kinase inhibitors	-Phase II trials ongoing
VEGFR		-Cabozantinib (multikinase-inhibitor that targets c-MET, VEGFR, RET)	-No overall improvement in the cabozantinib monotherapy arm -Combination with the ICI Atezolizumab achieved encouraging activity

CRPC: castration-resistant prostate cancer; IHC: immunohistochemistry; mHSPC: metastatic hormone-sensitive prostate cancer; ICI: Immune checkpoint inhibitor.

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
