# Peer review of "Targeted Approaches in Metastatic Castration-Resistant Prostate Cancer: Which Data?"

_cancers, 2022, doi:10.3390/cancers14174189_

Round 1

Reviewer 1 Report

This review summarized and discussed the new therapeutic approaches for mCRPC. This review emphasized the importance of gene sequencing in optimizing the selection of therapies for individual patients.

Moderate English changes are needed. 

Author Response

Response to Review 1: Thank you for your relevant suggestion. The paper has been revised improving the English, so we hope it now matches the journal standard.

Reviewer 2 Report

The current review aims to report an overview of recent progress in novel targeted therapies and provides a wide overview of pathways and the most common genomic aberrations in advanced prostate cancer. The manuscript is suitable for publication after a minor revision.

Comment: Studies have demonstrated that persistent AR signaling is the key driver in the progression to CRPC. In this scenario, cells acquire the ability to activate AR signaling either through AR gene amplification, AR mutation, constitutively active AR splice variants, or increased intratumor androgen production. Autophagy can be used by cancer cells to prolong their survival under harsh conditions of metabolic stress induced by various treatment modalities, such as castration therapy. Please consider including this reference (10.3390/ijms23073826); moreover, evidence suggests that autophagy itself may promote castration resistance. It has been demonstrated that the expression of autophagy regulation molecules (CAMKK2, AMPK, and ULK1) is related to the prognosis and progression of men with prostate cancer. 

Author Response

Response to Reviewer 2: Thank you for your comment. We updated the manuscript as suggested. In particular, we included information about the role of autophagy pathway in advanced prostate cancer in "Potential further novel agents for the management of metastatic castration-resistant prostate cancer" section of the manuscript at line 380-382 and we added the reference. 

Reviewer 3 Report

General Overview

Authors in the present manuscript have reviewed the recent progress and novel (current /potential) therapies for metastatic castration resistant prostate cancer. I found that the references are comprehensive and language is lucid. Clarity and context in this paper are good. This is much needed and timely review and may need minor revision before publication.

Minor comments:

1. Could mention about the CheckMate 650 Trial (NCT02985957) as a combinatorial [nivolumab (Opdivo) plus ipilimumab (Yervoy)] immunotherapy approach for mCRPC.

2. Could include the following information- Patients with CDK12 inactivation may benefit from immunotherapy (anti PD1 therapy). Clinical trial : NCT03570619.
Check the following reference for more information: Inactivation of CDK12 delineates a distinct immunogenic class of advanced prostate cancer; Cell, 173 (2018), 10.1016/j.cell.2018.04.034.

3. Abbreviations like ORR, HR, NHA and OS, used multiple times before providing the expanded form or what they stand for.

4. Few recent studies are not included.

Author Response

Response to Reviewer 3: Thank you for your observation. We updated the manuscript as suggested.

Point 1: Could mention about the CheckMate 650 Trial (NCT02985957) as a combinatorial [nivolumab (Opdivo) plus ipilimumab (Yervoy)] immunotherapy approach for mCRPC.

Response 1: We mentioned the CheckMate 650 at lines 296-300 and we added the reference.

Point 2: Could include the following information- Patients with CDK12 inactivation may benefit from immunotherapy (anti PD1 therapy). Clinical trial : NCT03570619. Check the following reference for more information: Inactivation of CDK12 delineates a distinct immunogenic class of advanced prostate cancer; Cell, 173 (2018), 10.1016/j.cell.2018.04.034

Response 2: We included the information suggested at line 301 and we added the reference

Point 3: Abbreviations like ORR, HR, NHA and OS, used multiple times before providing the expanded form or what they stand for.

Response 3: we updated the text as suggested

Point 4: Few recent studies are not included

Response 4: we included information about some recent studies in the text 

Round 2

Reviewer 1 Report

The manuscript can be accepted in present form. 

Reviewer 2 Report

The manuscript has been sufficiently improved to warrant publication in Cancers.

Reviewer 3 Report

Thank you so much for making the suggested changes.